# Real and Deepfake Face Recognition: An EEG Study on Cognitive and Emotive Implications

**DOI:** 10.3390/brainsci13091233

**Published:** 2023-08-23

**Authors:** Pietro Tarchi, Maria Chiara Lanini, Lorenzo Frassineti, Antonio Lanatà

**Affiliations:** 1Department of Information Engineering, University of Florence, 50139 Florence, Italy; pietro.tarchi@unifi.it (P.T.); mariachiara.lanini@unifi.it (M.C.L.); lorenzo.frassineti@unifi.it (L.F.); 2Department of Information Engineering, University of Pisa, 56122 Pisa, Italy

**Keywords:** face recognition, deepfakes, emotions, power spectrum, event-related potentials (ERPs)

## Abstract

The human brain’s role in face processing (FP) and decision making for social interactions depends on recognizing faces accurately. However, the prevalence of deepfakes, AI-generated images, poses challenges in discerning real from synthetic identities. This study investigated healthy individuals’ cognitive and emotional engagement in a visual discrimination task involving real and deepfake human faces expressing positive, negative, or neutral emotions. Electroencephalographic (EEG) data were collected from 23 healthy participants using a 21-channel dry-EEG headset; power spectrum and event-related potential (ERP) analyses were performed. Results revealed statistically significant activations in specific brain areas depending on the authenticity and emotional content of the stimuli. Power spectrum analysis highlighted a right-hemisphere predominance in theta, alpha, high-beta, and gamma bands for real faces, while deepfakes mainly affected the frontal and occipital areas in the delta band. ERP analysis hinted at the possibility of discriminating between real and synthetic faces, as N250 (200–300 ms after stimulus onset) peak latency decreased when observing real faces in the right frontal (LF) and left temporo-occipital (LTO) areas, but also within emotions, as P100 (90–140 ms) peak amplitude was found higher in the right temporo-occipital (RTO) area for happy faces with respect to neutral and sad ones.

## 1. Introduction

The human brain’s ability in face processing (FP) is crucial, as the decision on how to interact with other individuals mainly depends on the outcome of the recognition process [1]. Face recognition (FR) is a cognitive process whereby humans identify and discriminate between individuals based on facial features; its abilities vary across individuals and can be influenced by experience, familiarity, and attentional focus [2]. It involves activating and integrating specialized brain regions, such as the fusiform face area (FFA), which plays a key role in face processing [3]. Neural networks within the FFA extract relevant facial information, enabling the encoding and retrieval of facial representations from memory. The FFA, localized within the fusiform gyrus (FG), is responsible for extracting the consistent elements of facial features and their spatial arrangements that are inherently linked to a person’s identity. Specifically, the FFA is highly specialized in analyzing and discerning facial features, enabling the recognition and differentiation between individuals’ faces. The FFA showed increased activation when individuals perceive faces, in contrast to other objects or stimuli [4]. Faces can also be differentiated for their emotional content, and previous studies have largely investigated the recognition processes of face emotional expressions [5,6].

Early models of face perception, such as the influential model proposed by Bruce and Young [7], revealed distinct pathways for perceiving and recognizing different aspects of personal attributes from faces, including identity, emotion, and facial speech. According to these models, separate cognitive processes were involved, contributing to our overall understanding of FP. Observing emotional expressions on another person’s face allows us to gather information about the emotions they are experiencing and elicit similar emotional responses within ourselves. The perception of emotional expressions plays a vital role in social interactions and has been shown to activate brain regions involved not only in perceiving facial stimuli but also in experiencing one’s own emotions [5]. It suggests a close link between the perception of emotions in others and our own emotional experiences, highlighting the intricated relationship between social cognition and emotional processing. According to Adolphs [8], several brain regions involved in basic emotion recognition, including the temporo-occipital, orbito-frontal, and right parietal cortices, are engaged in processing the perceptual aspects and the emotional significance of stimuli. These regions are crucial in perceiving and analyzing the visual information of stimuli and assigning emotional value and significance to them. It suggests that these brain regions are involved in the integrated processing of perceptual and emotional aspects of stimuli.

Brain regions can be categorized as ‘affective’ or ‘cognitive’. However, it is evident that the brain regions typically associated with affective processes also play a role in cognition, and those considered cognitive regions also contribute to emotions. Noteworthy, cognition and emotion are strictly integrated and a complex cognitive–emotional dynamic exists among various brain networks. Emotion and cognition strongly influence behavior. Furthermore, the neural basis of emotion and cognition is highly interconnected and should be viewed as non-modular, with minimal capacity for decomposition [9].

In recent years, the rapid development of artificial intelligence (AI) has given rise to a concerning phenomenon known as deepfakes. Deepfakes are realistic digital media content, particularly images or videos, that portray false information and can be created from scratch or by modifying authentic content through deep learning algorithms. Media advanced creation technologies based on deep learning algorithms are universally acknowledged as a serious threat to a person’s reputation and digital identity, and, as the prevalence of deepfakes continues to grow, it becomes crucial to investigate their potential effects on human perception and cognition [10].

In the last decades, the human ability to distinguish between real and AI-generated faces has been investigated. Specifically, neuroscience research focused on how synthetic stimuli affect people’s capacity for face recognition tasks [11]. Multimedia forensic research investigated how much face-mixing operations (i.e., a face manipulation where two faces are mixed to create a hybrid one carrying traits of both original faces) are perceived by people. Ensuring restricted access to locations or services is of utmost importance, particularly in the context of face authentication systems, to prevent unauthorized entry [12]. Several studies [13,14,15,16] have reported that humans can still generally detect AI-generated media content correctly.

Recently, studies on electroencephalographic (EEG) correlates investigated viewers’ ability to distinguish familiar and unfamiliar people from their face-swapped counterparts. Results showed that it is possible to discriminate fake videos from genuine ones when at least one face-swapped actor is known to the observer [17].

Synthetic faces can also exhibit a broad range of emotions. Investigation of brain reactions to facial expressions is becoming a widespread research area, aiming to better understand emotional processing and cognitive mechanisms. Even though traditional models suggest that facial identity and expression are processed in distinctive brain areas, the current findings highlight that emotion processing can strongly influence facial recognition and memory mechanisms [18]. Finally, other studies have shown that FP in adults is modulated by the emotional relevance of faces, especially those with expressions of fear [19].

The main objective of this study is to explore the capacity of individuals to discriminate between real and fake faces generated by AI, with a particular focus on emotional expressions. This research is also motivated by the need to better understand the implications of deepfakes on human perception and the potential challenges they pose to distinguish between authentic and manipulated visual stimuli. Additionally, neural correlates of face processing and emotion recognition are investigated by analyzing electroencephalographic (EEG) data. Moreover, event-related potentials (ERPs) were evaluated as they offer a valuable understanding of the timing and neural mechanisms that underlie cognitive functions like perception, attention, and memory, with a high level of temporal precision.

The manuscript is organized as follows: Section 2 (Materials and Methods) will describe the experimental protocol, signal processing chain, and statistical analysis; Section 3 (Results) shows the results of power spectrum analysis (PSA) and ERP analysis. In Section 4 (Discussions), the obtained results and comparisons with current findings in the literature are reported. Finally, Section 5 (Conclusions) summarizes the results, limitations, and future research developments. Furthermore, the Appendix A, containing the set of Appendix A, reported all the 60 faces for the stimulation and the statistical analysis outcomes for the PSD-related feature for the emotional comparison.

## 2. Materials and Methods

### 2.1. Participants

In total, 23 healthy volunteers (13 F and 10 M; mean age—24.7 years, std age—2.8 years, median age—25 years, age range—19 to 29 years) were involved in the experimental session. All subjects reported normal or corrected-to-normal visual acuity. This study was approved by the Institutional Review Board and all participants gave written informed consent.

### 2.2. Experimental Protocol

Volunteers were subjected to 60 grayscale visual stimuli representing human faces, both synthetic or real, and expressing positive, neutral, or negative emotions. During the experimental session, room temperature (25 °C) and illumination condition (∼7800 lumen) were maintained constant; subjects were asked to sit on a chair, and the distance between the subject’s head and monitor was 60 cm. All sessions were conducted in the morning, from 9 am to 12 pm. All images were resized to have fixed dimensions (1024 pixel × 1024 pixel) and resolution (96 dpi) and were presented centered over a uniform background on the screen of a 24-inch full HD monitor. Real stimuli were selected from a set of Caucasian faces (age range of 20–50 years) included in the CK+ face database [20] by looking at the arousal and valence scores. The same face dataset was used for all participants, who were unfamiliar with the presented faces.

Synthetic faces were generated through a generative-AI algorithm (i.e., FaceMix) [21] by mixing together 4 grayscale real images, all expressing the same type of emotion, randomly sampled each time from the real faces set. Stimuli were presented only once, balanced in sex, type of emotional facial expression (positive, neutral, negative), and type of image, i.e., synthetic or real. The dataset comprised three classes, specifically, the synthetic class, real class, and emotional class, where the latter included both real and synthetic faces split for the 3 different emotional expressions. Volunteers sat on a chair wearing an EEG helmet in front of a monitor where stimuli were presented. The experimental protocol was composed of two phases, as shown in Figure 1. The first phase was a 4 min baseline acquisition with 2 min of closed and 2 min of open eyes. In the second phase, subjects observed 60 images of faces (for a complete overview, please refer to Appendix A), composed by 3 sets of 20 faces (10 real and 10 synthetic). Each set contained faces associated exclusively with a polarized mood: positive (happy or smiling faces, Figure 2a,b and Appendix A), referred to as “happy”; neutral (relaxed faces, neutral expressions, Figure 2c,d and Appendix A); or negative (sad, angry or discomforted faces, Figure 2e,f and Appendix A), referred to as “sad”. Both faces and sets were presented randomly for each subject. Each face was observed for 10 seconds by the subject, who was requested to finally press “z” or “m” on a keyboard if the presented stimuli were, respectively, considered synthetic or real. Image presentation and response times were, respectively, managed and recorded through the software interface.

EEG monitoring and acquisition were performed through the DSI-24 helmet (Wearable Sensing, San Diego, CA, USA) with Ag/AgCl dry electrodes. The helmet consists of 21 electrodes (19 EEG channels, Pz channel is used as a common reference, and two auricular electrodes), was arranged according to the International 10–20 Standard, and was wirelessly connected to a triggering hub device for neurophysiological signal synchronization. Specifically, the trigger allowed us to synchronize the EEG data and presented stimuli through a photodiode applied to the screen. EEG and trigger data were collected at a sampling frequency of 300 Hz. The list of EEG channels used is given below: Fp1, Fp2, F3, F4, F7, F8, Fz, C3, C4, Cz, T3, T4, T5, T6, P3, P4, O1, O2.

### 2.3. Signal Processing Chain

EEG data were analyzed in MATLAB environment through EEGLAB [22] for continuous and event-related EEG processing. EEG signals were pre-processed following the standardized Harvard Automated Processing Pipeline for Electroencephalography (HAPPE) [23]. EEG pre-processing (Figure 3) included the following steps:Band-pass filtering (1–45 Hz);Channel selection;50 Hz electrical noise removal through Cleanline EEGLAB plugin;Crude bad channel detection using spectrum criteria and 3 standard deviations as channel outlier threshold;Independent component analysis (ICA) for clustering the data;Wavelet-enhanced independent component analysis (W-ICA) for thresholding with a level 5 coiflet wavelet and threshold multiplier 0.75;Multiple artifact rejection algorithm (MARA) for independent component rejection if artifact probability is greater than 0.5;Segmentation in epochs of 10 s each;Interpolation of bad data within segments from good channels only;Rejection of bad segments using amplitude-based and joint probability artifact detection;Channel interpolation with the spherical method;Average re-referencing of channels.

After the pre-processing phase, EEG signals were split into epochs of 10 s identifying each precise stimulus. Of the total 1360 extracted epochs (Table 1), 1224 were retained for further analysis (Table 2), while 136 were excluded by segmentation and rejection processes.

### 2.4. Statistical Analysis

For each epoch, the average power spectrum in the 6 frequency bandwidths of analysis (i.e., delta, 1–4 Hz; theta, 4–7 Hz; alpha, 8–12 Hz; low-beta, 13–17 Hz; high-beta, 18–32 Hz; gamma, 32–48 Hz) was computed with a 300-point fast Fourier transform (FFT) without overlap. PSD-related features were computed as the relative power densities for each band by normalizing the mean power spectra of each band by the power spectrum mean value in the 1–48 Hz range [24].

The event-related potential (ERPs) analysis was conducted by grand averaging the EEG signal in the first 500 ms post-stimulus without performing baseline correction [25]. N100 (80–120 ms) [26], P100 (90–140 ms) [27], face-specific N170 (140–210 ms) [28], and N250 (200–300 ms) [29] and early P300 (300–400) [28] components were investigated in left temporo-occiptal (LTO, includes T3, T5, and O1 channels), right temporo-occipital (RTO, includes T4, T6, and O2 channels), left frontal (LF, includes F3 and F7 channels), and right frontal (RF, includes F4 and F8 channels) areas. For each ERP component, the peak and its latency with respect to the onset of the stimulus were computed for every subject.

Epochs were labeled as follows to facilitate further analysis:Happy—in these epochs, the presented face expressed a “happy” emotion;Neutral—in these epochs, the presented face expressed a neutral emotion;Sad—in these epochs, the presented face expressed a “sad” emotion;tt (i.e., true–true)—in these epochs, the presented face was real, and the subject’s answer was “real”;ff (i.e., false–false)—in these epochs, the presented face was synthetic, and the subject’s answer was “synthetic”;tf (i.e., true–false)—in these epochs, the presented face was real, and the subject’s answer was “synthetic”;ft (i.e., false–true)—in these epochs, the presented face was synthetic, and the subjects’ answer was “real”.

Since the EEG-related PSD-extracted features were not normally distributed according to the Shapiro–Wilk test, surrogate tests were performed for statistical analysis [30]. The bootstrap method was performed to estimate statistics by sampling our dataset with replacement. After performing bootstrap statistics, a paired *t*-test was carried out to verify whether the mean values of the parameters were statistically different at a significance level of 95% (*p* < 0.05). For emotional comparisons, a post hoc Bonferroni correction was performed.

As for the ERP statistical analysis, the Shapiro–Wilk test was also performed on peak and peak latency features. If both conditions under comparison were normally distributed, a paired *t*-test was used, or the Wilcoxon rank-sum test was performed. For emotional comparisons, a post hoc Tukey–Kramer correction was performed. Statistical comparison tests were organized as follows:Imfalse vs. Imtrue: comparison between synthetic and real stimulation.-HappyF vs. HappyT: comparison between synthetic happy and real happy stimulation.-NeutralF vs. NeutralT: comparison between synthetic neutral and real neutral stimulation.-SadF vs. SadT: comparison between synthetic sad and real sad stimulation.Happy vs. Neutral: comparison between happy vs. neutral emotions-Imfalse_H vs. Imfalse_N: comparison between synthetic happy vs. synthetic neutral emotions.-Imtrue_H vs. Imtrue_N: comparison between real happy vs. real neutral emotionsHappy vs. Sad: comparison between happy vs. sad emotions-Imfalse_H vs. Imfalse_S: comparison between synthetic happy vs. synthetic sad emotions-Imtrue_H vs. Imtrue_S: comparison between real happy vs. real sad emotionsNeutral vs. Sad: comparison between neutral vs. sad emotions-Imfalse_N vs. Imfalse_S: comparison between synthetic neutral vs. synthetic sad emotions-Imtrue_N vs. Imtrue_S: comparison between real neutral vs. real sad emotions

## 3. Results

This section reports on the statistical results regarding PSD and ERP features. In the first subsection, results of power spectrum analysis delta, theta, alpha, low-beta, high-beta, and gamma bands are shown, while in the second subsection, the results of the ERP analysis are reported.

### 3.1. Power Spectrum Analysis

Power spectrum statistical analysis is reported through scalp topographic maps (STMs). STMs describe the spatial distribution of extracted parameters, computed at the electrode position, across the brain. To simplify visualization, we used a false-colors STM (FCSTM) highlighting the statistically significant areas (*p* < 0.05, *p* < 0.01 and *p* < 0.001). Non-significant areas were standardized with the color grey. The FCSTM represents the *p*-value of the paired *t*-test comparing the averages of the two conditions under investigation. If an area of the FCSTM assumes warm colors (yellow, orange, red), the first term of the comparison is statistically greater than the second one; on the contrary, if the area assumes cold colors (cyan, light blue, blue) the second term is statistically greater than the first.

Moreover, in this study we proposed a set of stimuli that integrated cognitive and emotional processes. This complex set of stimuli required a specific investigation on the brain dynamic and the temporal resolution of FR process. Therefore, epoch analysis was performed on three different time intervals: 0–5 s, 5–10 s, and 0–10 s, allowing an interesting comparison among the first half (0–5 s), the second half (5–10 s), and the entire epoch (0–10 s) durations.

#### 3.1.1. Imfalse vs. Imtrue

As for the “Imfalse vs. Imtrue” comparison, results of statistical analysis, as reported by Figure 4 (0–5 s), Figure 5 (5–10 s), and Figure 6 (0–10 s), show significant delta activation for deepfakes in the frontal (Figure 5a and Figure 6a) and right occipital areas (Figure 4a, Figure 5a and Figure 6a); also, the left temporal area (Figure 5e and Figure 6e) shows significant high-beta activations for deepfakes, whereas significant turn-ons for real faces are shown in theta in the right frontal area (Figure 4b and Figure 6b), in alpha in the right occipital (Figure 4c and Figure 6c) and left central areas (Figure 4c, Figure 5c and Figure 6c), and in high-beta and gamma in the right parietal area (Figure 5e,f and Figure 6e,f).

#### 3.1.2. Emotional Comparison

As for the emotional comparison, the results of significant EEG activation are briefly reported in Figure 7 (0–10 s). For a complete overview of the results regarding emotional comparison, please refer to Appendix A.

In the “Happy vs. Neutral” comparison, greater significant activation was in the delta band in the frontal and occipital areas (Figure 7a, Appendix A) and in the alpha band in the left temporal and right parietal areas (Figure 7c and Appendix A) for faces expressing neutral emotions.In the “Happy vs. Sad” comparison, greater significant activities were in the theta band in pre-frontal and left occipital areas (Figure 7h and Appendix A) and in the low-beta band in the frontal and right occipital areas (Figure 7j and Appendix A) for faces expressing positive emotions, whereas there was greater significant activation in the alpha band in the left temporal area (Figure 7i and Appendix A) and in high-beta band in the right frontal area (Figure 7k and Appendix A) for faces expressing negative emotions. It is worth noting that in the first 5 seconds (Appendix A), faces expressing negative emotions elicited more significant activation, whereas in the last 5 s of the epoch (Appendix A), faces expressing positive emotion were predominant in determining statistical significance.In the “Neutral vs. Sad” comparison, significant activations in the low-beta band were found in the right occipital and left temporal areas (Figure 7p, Appendix A) for faces expressing neutral emotions.

### 3.2. ERP Analysis

ERP analysis was performed considering four areas of the cerebral cortex: LF, RF, LTO, and RTO. This was due to the different roles which these areas serve in face processing (LTO and RTO, as reported by Barton [31]) and decision making (LF and RF, as reported by Collins et al. [32]).

#### 3.2.1. Originality Comparisons

Statistical analysis on ERP components for originality comparison provided the following results (Table 3 and Figure 8 and Figure 9).

#### 3.2.2. Emotional Comparison

As for the statistical analysis on ERP components for emotional comparisons, this produced the following results (Table 4).

## 4. Discussion

This research aimed to investigate how people could distinguish between real and AI-generated faces emphasizing emotional expressions and comprehend the effects of deepfakes on human perception and the possible difficulties they present in differentiating between real and artificial visual stimuli. To this purpose, a statistical analysis of EEG correlates in terms of PSD- and ERP-related features has been performed.

On the behavioral level, a good deepfake discrimination capacity has been found, which confirmed studies on the recognition of AI-generated faces [13,14,15,16]. Participants had good performance in recognizing the true faces as well [33,34]. A good degree (∼76%) of accuracy in classifying faces was observed, as reported in Table 1 and Table 2. Participants were slightly better at discriminating images with neutral emotional content than images with positive or negative emotional content. It seems to confirm the work of Montagrin et al., which highlighted the important role played by memory in facilitating the recognition of neutral faces in goal-relevant situations [35]. Statistical analysis of PSD-related features highlighted two main significant turn-ons for deepfakes: delta activation in the frontal (Figure 5a and Figure 6a) and right occipital (Figure 4a, Figure 5a and Figure 6a) areas could be due to a dynamic switching attention mechanism [36], meaning that participants spent more time interpreting synthetic faces, whereas high-beta activation in the left temporal area, which includes the FG [37], states that FFA activation is not determined by the originality label of the face presented.

Theta activations in real faces were observed in the right frontal area (Figure 4b, Figure 5b and Figure 6b), according to Canales et al. [38], it indicated an increase in short-range frontal theta synchronization associated with visual imagery of faces and also with the need for cognitive control [39]. Theta-, alpha-, high-beta-, and gamma-significant activities for real faces were all over the right hemisphere (RH), as shown in Figure 4b,c,e,f, Figure 5b,c,e,f and Figure 6b,c,e,f, might instead hint a distinct pathway in the brain to discriminate real faces from synthetic ones, which is in line with findings by Sergent et al., suggesting a right hemisphere predominance for real faces when compared to objects [40]. It has been linked to increased left visual field (LVF) activity during FR tasks [41].

Our findings in the emotional comparison show that it is feasible to discriminate between positive, negative, and neutral emotions. Results of frontal theta activation for happy faces (Figure 7h), especially in the second part of the epoch (Appendix A), are consistent with findings by Knyazev et al. reporting higher sensitivity to happy faces than to angry ones in the late, conscious FP stage [42]. Moreover, other statistical outcomes are reported in the Appendix A and show how brain activation drastically changes between the first and the latter 5 seconds of the epoch in the “Happy vs. Sad” comparison more than any other.

Current ERP research findings have shown interesting results in FR processing. Specifically, fearful facial expressions have been found to produce a more pronounced negative N100 component than happy and neutral faces [43]. In contrast, the P100 component has shown to be sensitive mainly to domain-general visual processes and can be considered as a marker of individual face recognition [44] or, at least, of category-level face processing [45,46]. The N170 component has been proved to be face-specific with respect to most objects, and increasing its latency when the structure of a face is hard to perceive [47]; the N250 component is, instead, thought to be generated in or near the FFA: it increases if a face image is the same as an immediately preceding face as compared with when it is different [47]. Finally, research on the early P300 has hinted that such components may reflect categorization and attention to motivational, relevant information, including emotion, gender, or identity [48]. Our results on the ERP statistical analysis, while not finding evidence in the literature, still suggest that brain activity might process AI-generated and real faces differently. In fact, in the comparison “Imfalse vs. Imtrue” (Table 3), the N250 component presents a lower latency peak for real faces both in the RF and LTO areas (Figure 8b,c), whereas earlier components seem to be modulated, both in peak amplitude (P100) and latency (N100, N170), by deepfakes (Table 3). As for emotional comparison, an increase in peak amplitude of the P100 component in RTO for happy faces compared to neutral and sad ones (Table 4, Figure 10 and Figure 11) was found consistently. In contrast, the “Neutral vs. Sad” comparison did not produce any significant outcomes until the split in deepfakes (N100 peak amplitude increased in the neutral class) and real (N170 peak amplitude increased in the sad class) faces.

During preliminary analysis, we reported no significant difference in the outcomes of the labeling task between female and male participants before the pre-processing phase. Significance was found solely on the labeling outcomes of the remaining epochs after pre-processing for the “sad_tf” labeled faces (*p*-value = 0.0346), with female participants being more inclined to classify as “synthetic” a real face expressing negative emotion. Moreover, considering the small sample size, we did not investigate gender differences for power spectrum and ERP analyses.

The present study contributes to the deepfake perception, FP, and emotional engagement literature. The main point was the interplay of cognitive and emotional processes. Engaging these two sides helped us build a solid starting point for future research and further analysis involving different emotional and cognitive tasks. Similar studies could investigate behavioral and neurophysiological correlates in fragile subjects, such as older adults or persons with psychological diseases, to better understand distinct patterns in their neurophysiological response, aiming to provide an early evaluation of healthy elderly cognitive status. Future studies might also explore how one’s affect influences the judgment on emotional content about external context. Recent research shows that some emotions can diminish, for example, deceit [49,50]. It is worthwhile noting that the proposed study presents some limitations. The first regards the small sample size of the involved subjects, which also precluded us from a gender difference investigation. Another limitation is the pre-processing pipeline, which can lead to different outcomes. Other EEG pre-processing pipelines, such as PREP [51] and APP [52], should be investigated in future works for results robustness and repeatability assessment. Finally, FP dynamic investigation is an open question that deserves an appropriate deepening, since integrating cognitive and emotional brain processes activates complex brain responses regarding behavior and interaction among brain areas and networks. The future directions include increasing the number of participants to achieve more robust and accurate outcomes, as well as using self-report questionnaires from participants to pair a possible recognition strategy adopted by participants during the experiment with the results of the EEG analysis. Moreover, it could enable the application of more sophisticated modeling methodologies to gain deeper insights into cognitive and emotional engagement in perceiving and discriminating between real and synthetic faces.

## 5. Conclusions

This work aimed to investigate the EEG correlates of a healthy group subjected to a visual task with cognitive and emotional implications to discriminate significant brain activations in the different proposed stimulating cases. The novelty of the work concerned the comparison between real and synthetic faces, for which, to the best of our knowledge, no previous work has been found in the literature comparing the two conditions and how their emotional content could modulate the participants’ brain activation. Despite some obvious limitations, such as the small number of subjects studied and the lack of knowledge and references on the FP time dynamics for power spectrum analysis, the reported findings have led to many open questions that deserve to be explored in future works, especially regarding the characterization of neurophysiological dynamics during emotional deepfake discrimination tasks as well as in terms of gender differences.

## Figures and Tables

**Figure 1 brainsci-13-01233-f001:**
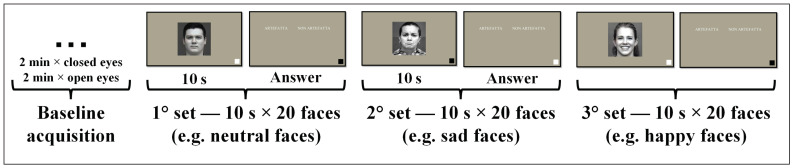
Experimental protocol timeline.

**Figure 2 brainsci-13-01233-f002:**
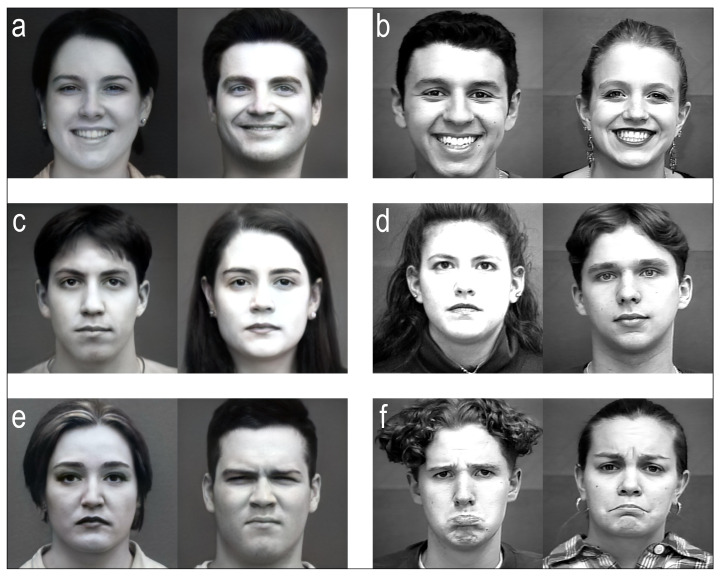
Example of synthetic (**a**,**c**,**e**) and real (**b**,**d**,**f**) faces expressing positive (**a**,**b**), neutral (**c**,**d**), and negative emotions (**e**,**f**) used as stimuli.

**Figure 3 brainsci-13-01233-f003:**
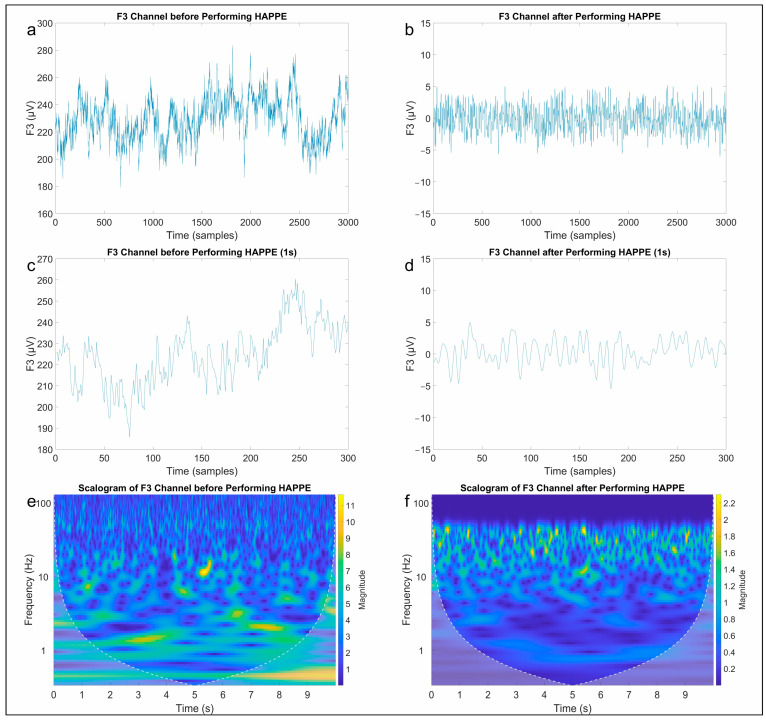
Example of EEG data from P3 channel before (**a**,**c**,**e**) and after (**b**,**d**,**f**) performing HAPPE.

**Figure 4 brainsci-13-01233-f004:**
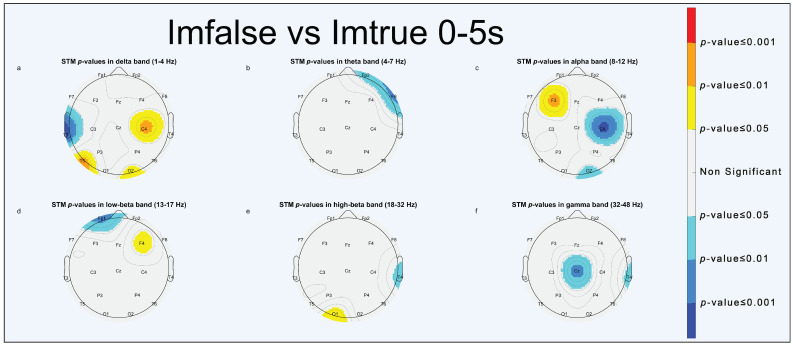
STMs of significant statistical activation of brain areas for “Imfalse vs. Imtrue” comparison (0–5 s). (**a**) delta band, (**b**) theta band, (**c**) alpha band, (**d**) low-beta band, (**e**) high-beta band, (**f**) gamma band.

**Figure 5 brainsci-13-01233-f005:**
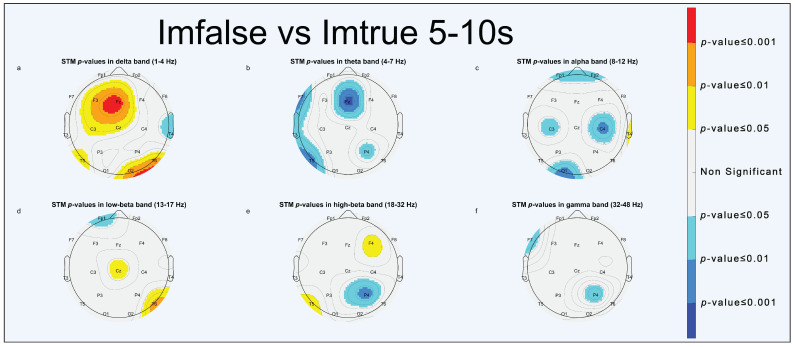
STMs of significant statistical activation of brain areas for “Imfalse vs. Imtrue” comparison (5–10 s). (**a**) delta band, (**b**) theta band, (**c**) alpha band, (**d**) low-beta band, (**e**) high-beta band, (**f**) gamma band.

**Figure 6 brainsci-13-01233-f006:**
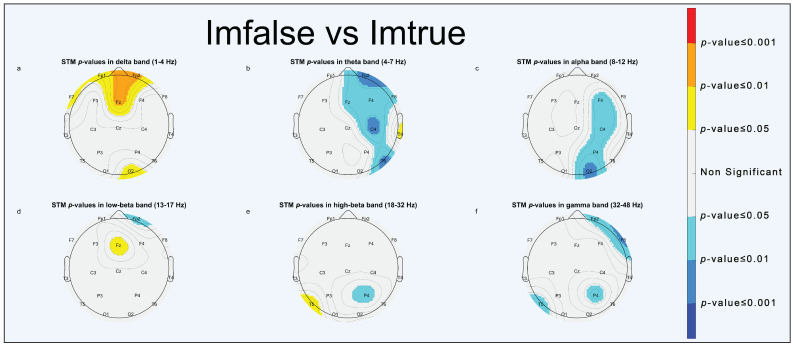
STMs of significant statistical activation of brain areas for “Imfalse vs. Imtrue” comparison (0–10 s). (**a**) delta band, (**b**) theta band, (**c**) alpha band, (**d**) low-beta band, (**e**) high-beta band, (**f**) gamma band.

**Figure 7 brainsci-13-01233-f007:**
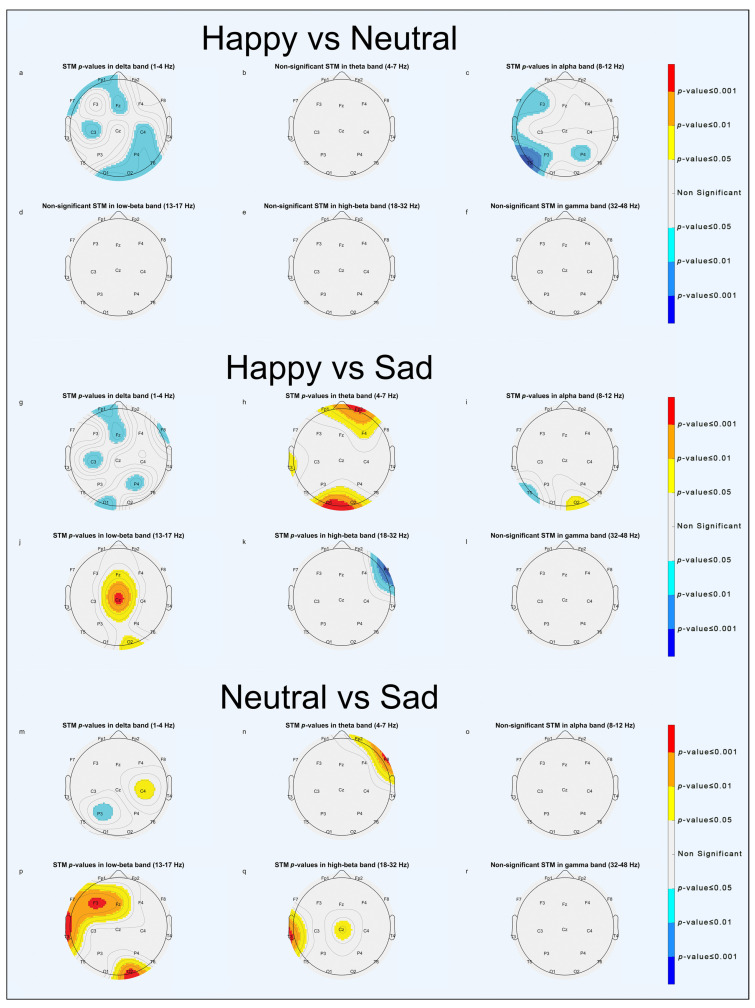
STMs of significant statistical activation of brain areas in emotional comparisons (0–10 s). “Happy vs. Neutral” comparison (**a**–**f**): (**a**) delta band, (**b**) theta band, (**c**) alpha band, (**d**) low-beta band, (**e**) high-beta band, (**f**) gamma band. “Happy vs. Sad” comparison (**g**–**l**): (**g**) delta band, (**h**) theta band, (**i**) alpha band, (**j**) low-beta band, (**k**) high-beta band, (**l**) gamma band. “Neutral vs. Sad” comparison (**m**–**r**): (**m**) delta band, (**n**) theta band, (**o**) alpha band, (**p**) low-beta band, (**q**) high-beta band, (**r**) gamma band.

**Figure 8 brainsci-13-01233-f008:**
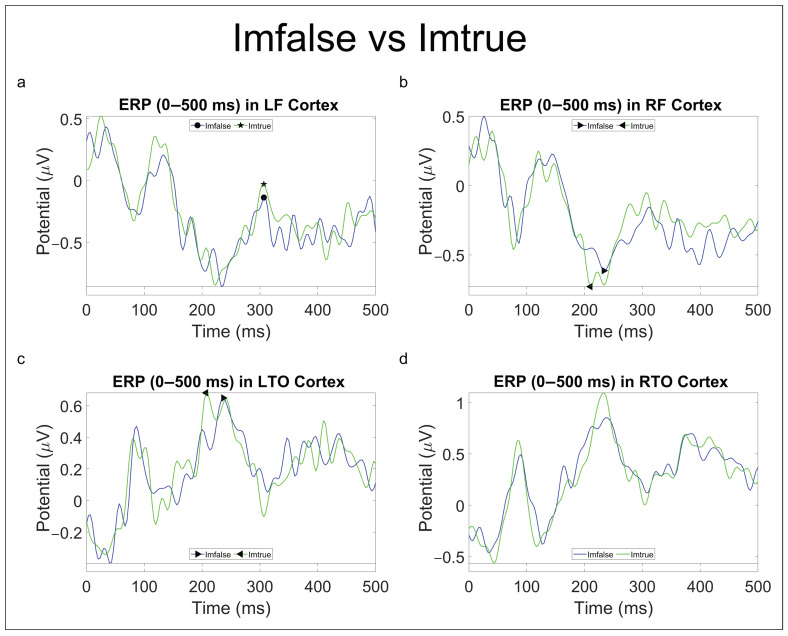
ERP (0–500 ms) in LF (**a**), RF (**b**), LTO (**c**), and RTO (**d**) areas for “Imfalse vs. Imtrue” comparison. Significant *p*-values (p<0.05) for peaks amplitude are reported with the ★ (indicating prevalence) and ● symbols, whereas for peaks latency, they are reported with the ◂ (indicating prevalence) and ▸ symbols.

**Figure 9 brainsci-13-01233-f009:**
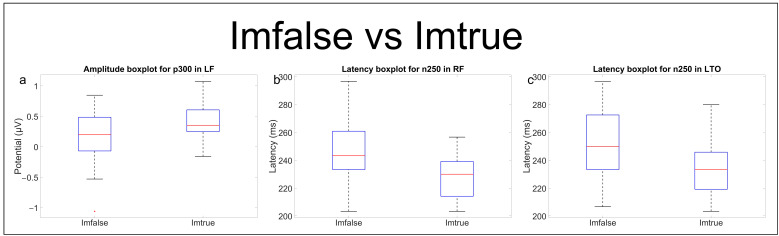
Boxplots of amplitude distributions in LF area (**a**) for P300 component and latency distributions for N250 component in RF (**b**) and LTO (**c**) areas for “Imfalse vs. Imtrue” comparison.

**Figure 10 brainsci-13-01233-f010:**
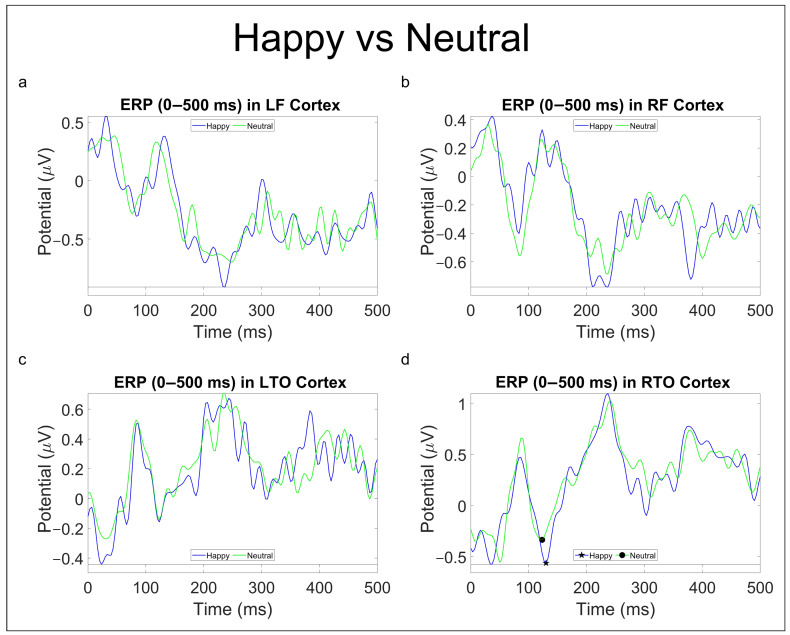
ERP (0–500 ms) in LF (**a**), RF (**b**), LTO (**c**), and RTO (**d**) areas for “Happy vs. Neutral” comparison. Significant *p*-values (p<0.05) for peak amplitude are reported with the ★ (indicating prevalence) and • symbols.

**Figure 11 brainsci-13-01233-f011:**
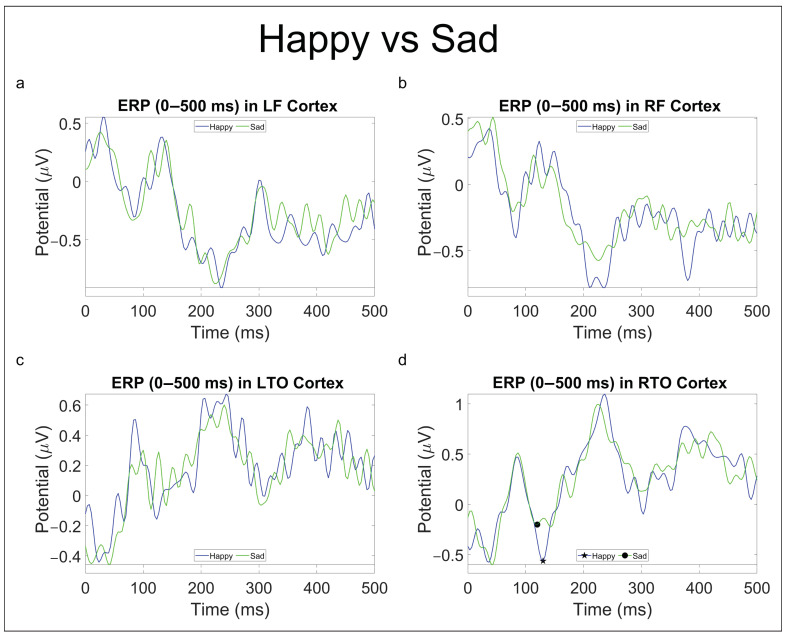
ERP (0–500 ms) in LF (**a**), RF (**b**), LTO (**c**), and RTO (**d**) areas for “Happy vs. Sad” comparison. Significant *p*-values (p<0.05) for peak amplitude are reported with the ★ (indicating prevalence) and • symbols.

**Table 1 brainsci-13-01233-t001:** Epochs division according to labels before HAPPE.

Epochs Label	tt	ff	tf	ft	Total	Percentage
happy	173	172	57	58	460	33.3%
neutral	180	181	50	49	460	33.3%
sad	171	172	59	58	460	33.3%
total	524	525	166	165	1380	100%
percentage	38%	38%	12%	12%	100%	

**Table 2 brainsci-13-01233-t002:** Epochs division according to labels after HAPPE.

Epochs Label	tt	ff	tf	ft	Total	Percentage
happy	148	151	51	45	395	32.3%
neutral	159	162	45	43	409	33.4%
sad	158	153	56	53	420	34.3%
total	465	466	152	141	1224	100%
percentage	38%	38.1%	12.4%	11.5%	100%	

**Table 3 brainsci-13-01233-t003:** Statistical analysis results on ERP components peak and peak latency in originality comparisons (deepfakes vs. real faces). Prevalence is reported in brackets; “F” stands for false and “T” for true.

Comparisons	LF	RF	LTO	RTO
Imfalse vs. Imtrue	P300 (T) ↑	N250 (T) ←	N250 (T) ←	-
HappyF vs. HappyT	N100 (F) ←	-	P300 (T) ↑	-
NeutralF vs. NeutralT	-	-	-	N170 (F) ←
SadF vs. SadT	N250 (T) ←	-	P100 (F) ↑	-

↑ = significance in peak amplitude, ← = significance in peak latency.

**Table 4 brainsci-13-01233-t004:** Results of statistical analysis on ERP components peak and peak latency for emotional comparisons (multiple comparisons between happy, neutral, and sad faces). Prevalence is reported in brackets; “H” stands for happy, “N” for neutral, and “S” for sad.

Comparisons	LF	RTO
Happy vs. Neutral	-	P100 (H) ↑
Happy vs. Sad	-	P100 (H) ↑
Imfalse_N vs. Imfalse_S	N100 (N) ↑	-
Imtrue_N vs. Imtrue_S	N170 (S) ↑	-

↑ = significance in peak amplitude.

## Data Availability

The data presented in this study are available on request from the corresponding author. The data are not publicly available due to privacy reasons.

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
