# Peer review of "Real and Deepfake Face Recognition: An EEG Study on Cognitive and Emotive Implications"

_brainsci, 2023, doi:10.3390/brainsci13091233_

Round 1

Reviewer 1 Report

Thank you for the submission of the study. The authors employed EEG to examine the discriminative capacity of individuals in distinguishing between authentic and Deepfake facial representations, encompassing emotional expressions of happiness, sadness, and neutrality. While the chosen research direction is undeniably intriguing, it is imperative to address specific pivotal aspects in order to enhance the scientific rigor and applicability of the work.

  • The resolution of Figure 1 should be enhanced for improved visual clarity and detail.

  • Figure 2 would benefit from the inclusion of additional examples of synthetic and authentic faces..

  • In Figures 3-10, the axis labels and tick values appear to be illegible. To enhance readability, it is recommended to employ bigger font sizes.

  • In Figures 3-6, "d" has been mistyped as "a".

  • In the context of the STM p-value heat maps (Figures 4-6), it is important to provide more details about the findings in terms of the time intervals. It is essential to elucidate how the results for the 0-5 seconds and 5-10 seconds intervals align with the findings for the 0-10 second duration.

  • The ERP plots fail to indicate the presence of errors, and caution should be exercised in relying solely on p-values for interpretation due to the limited sample size.

  • It would be intriguing to observe the data split by gender.

  • No effective controls were shown in this work.

Author Response

Answer to Comments – Rev #1

  • The resolution of Figure 1 should be enhanced for improved visual clarity and detail.
    • Thank you for your comment. We provided to enhance the visual clarity of Figure 1 as suggested.
  • Figure 2 would benefit from the inclusion of additional examples of synthetic and authentic faces.
    • Thank you for the suggestion. We added a new figure (Figure S1) in Supplementary Materials, including all 60 faces used during the experiment. The manuscript has been changed accordingly. See changes reported in Section 2 (Methods and Materials), line 131.
  • In Figures 3-10, the axis labels and tick values appear to be illegible. To enhance readability, it is recommended to employ bigger font sizes.
    • Thank you for your advice. Axis labels and tick values for Figures 3-11 (we included a new figure in the paper, the Figures previously labeled 9-10 have been changed to Figures 10-11) have been increased to enhance readability.
  • In Figures 3-6, "d" has been mistyped as "a".
    • Thank you for your observation. We corrected the typos in the captions for Figures 3-6.
  • In the context of the STM p-value heat maps (Figures 4-6), it is important to provide more details about the findings in terms of the time intervals. It is essential to elucidate how the results for the 0-5 seconds and 5-10 seconds intervals align with the findings for the 0-10 second duration.
    • Thank you for your comment. Unfortunately, we didn't find any reference for the timing proposed, as face recognition timing has been studied in the early stages after the onset of the visual stimuli (maximum 1 s after onset). We decided to split the epoch into two parts of equal length to get a first clue about the temporal resolution of the face recognition process, which, in the proposed experiment, integrated cognitive and emotional processes and therefore required an extended analysis window for the EEG PSD-related features. Also, it allowed us to compare the outcomes of the first (0-5 s) and the second half (5-10 s) of each epoch with the ones of the entire epoch (0-10 s). See changes reported in Section 3 (Results), lines 246-250, and Section 4 (Discussions), lines 368-371.
  • The ERP plots fail to indicate the presence of errors, and caution should be exercised in relying solely on p-values for interpretation due to the limited sample size.
    • Thank you for your observation. In Section 3 (Results), we managed to add a new figure (Figure 9), which reports boxplots regarding the significance previously reported in Figure 8.
  • It would be intriguing to observe the data split by gender.
    • Thank you for your suggestion. However, considering the small sample size, we did not further investigate the statistical comparison tests in terms of gender differences both for power spectrum and ERP analysis (see changes in the Discussion section, lines 347-353). In future studies, with a more consistent sample size of participants, this investigation could provide some interesting evidence.
  • No effective controls were shown in this work.
    • Thank you for your comment. We provided more details about the experimental protocol, showing all the effective controls. See changes reported in Section 2 (Methods and Materials), lines 111-115.

Reviewer 2 Report

This is a very valuable research. As we know, with the development of deep learning applications in artificial intelligence, machine learning models can generate non-existent faces that can even have rich expressions. This technology is very attractive, but it may also be used for some illegal purposes. The research results of this paper can be used to detect machine generated fake faces, even if these faces can have different expressions. The author has confirmed through experiments that the proposed method can accurately distinguish between real facial expressions and fake facial expressions. I believe that the research in this paper has high value, scientific methods, reasonable data, and clear conclusions, making it a relatively excellent research. I have two small questions here for reference only.

1. The amount of subjects can be increased in future research to make the research conclusions more convincing.

2. Some limitations of this study can be listed in the paper, such as how noise removal is performed when obtaining EEG data.

Author Response

Answer to Comments – Rev #2

  • The amount of subjects can be increased in future research to make the research conclusions more convincing.
    • Thank you for your suggestion. Surely that would be the case, and we plan to do it in future research studies.
  • Some limitations of this study can be listed in the paper, such as how noise removal is performed when obtaining EEG data.
    • Thank you for your comment. We managed to add more details about the pre-processing pipeline (see Methods and Materials Section, lines 154-169). Also, we acknowledged that the choice of a pre-processing pipeline rather than another, along with all the parameters of the steps performed, can lead to different outcomes: other EEG pre-processing pipelines should be investigated in future works to ensure that the obtained results are reproducible. See changes reported in Section 4 (Discussions), lines 365-368.

Reviewer 3 Report

The discussions and  conclusions   might  be improved i.e. by highlighting the novelty proposed in this paper

Author Response

Answer to Comments – Rev #3

  • The discussions and conclusions might be improved i.e. by highlighting the novelty proposed in this paper
    • Thank you for your comment. Following your suggestion, we added the Conclusion section (Section 5) and highlighted the novelty proposed by our study. See changes reported in Section 5 (Conclusions), lines 382-286.

Reviewer 4 Report

This article is devoted to the study of distinguishing between real and fake human faces expressing positive, negative or neutral emotions. The manuscript describes the experimental protocol, signal processing chain and statistical analysis, as well as the results of EEG power spectrum analysis and analysis of event-related potentials. The manuscript is clear, relevant to the field, and presented in a well-structured manner.

The links provided are mostly recent publications and are current.

The conclusions of the manuscript are not scientifically substantiated. The study sample is small. There is no data on self-reports of subjects indicating the reasons for their separation of real and deepfake faces. The manuscript does not describe strategies for such separation. The manuscript lacks data on the differences between men and women in recognizing artificial and real faces.

It is difficult to conclude how reproducible the results of the manuscript are based on the details provided in the methods section.

Ethics statements and data availability statements are adequate.

Author Response

Answer to Comments – Rev #4

  • The study sample is small.
    • Thank you for your comment. Surely that is the case, and we plan to increase it in future research studies.
  • There is no data on self-reports of subjects indicating the reasons for their separation of real and deepfake faces. The manuscript does not describe strategies for such separation.
    • Thank you for your observation. Unfortunately, we did not propose a self-report questionnaire to participants as we focused mainly on the outcomes of the physiological signal analysis. That is an interesting point of inspiration for future studies, in which we may be able to pair the strategy adopted by them during the experiment with the results of the EEG analysis. See changes reported in Section 4 (discussion), lines 372-374.
  • The manuscript lacks data on the differences between men and women in recognizing artificial and real faces.
    • Thank you for your suggestion. Considering the small sample size, we did not further investigate the statistical comparison tests regarding gender differences for power spectrum and ERP analysis. However, we reported gender differences in labelling task outcomes (see changes in the Discussion section, lines 347-353). In future studies, with a more consistent sample size of participants, this investigation could provide some interesting evidence.
  • It is difficult to conclude how reproducible the results of the manuscript are based on the details provided in the methods section.
    • Thank you for your comment. Following your suggestion, we added more details about the pre-processing pipeline to improve the reproducibility of the results. See changes reported in Section 2 (Material and Methods), lines 382-154-169.

Round 2

Reviewer 1 Report

Thanks to the authors for the changes. 

Reviewer 4 Report

Thank you for you comments.